# Hydrologic Trends in the Upper Nueces River Basin of Texas—Implications for Water Resource Management and Ecological Health

**E. Dave Thomas [1], Kartik Venkataraman [2],*, Victoria Chraibi [3] and Narayanan Kannan [4]** 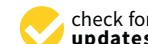

[1] Department of Chemistry, Geosciences, and Physics, Tarleton State University, Stephenville, TX 76402, USA; ermon.thomas@gmail.com

[2] Department of Engineering and Computer Science, Tarleton State University, Stephenville, TX 76402, USA

[3] Department of Biological Sciences, Tarleton State University, Stephenville, TX 76402, USA; chraibi@tarleton.edu

[4] Texas Institute for Applied Environmental Research, Tarleton State University, Stephenville, TX 76402, USA; kannan@tarleton.edu

\* Correspondence: venkataraman@tarleton.edu; Tel.: +1-254-968-9164

**Abstract:** Reliable water sources are central to human and environmental health. In south Texas, USA, the Nueces River Basin (NRB) directly or indirectly plays that important role for many counties. Several NRB stream segments are designated as ecologically significant because they serve crucial hydrologic, ecologic, and biologic functions. The hydrologically significant streams recharge the Edwards Aquifer, an essential water source for the region's agricultural, industrial, and residential activities. Unfortunately, the semiarid to arid south Texas climate leads to large inter-annual precipitation variability which impacts streamflow, and as a consequence, the aquifer's recharge. In this study, we used a suite of hydrologic metrics to evaluate the NRB's hydroclimatic trends and assess their potential impacts on the watershed's ecologically significant stream segments using precipitation and streamflow data from the National Climatic Data Center (NCDC) and Hydroclimatic Data Network (HCDN) respectively from 1970 to 2014. The results consistently showed statistically significant decreasing streamflow for certain low-flow indicators over various temporal scales, likely due to water rights diversions and minimal land use changes. This research could help decision-makers develop the necessary tools to manage water resources in south Texas, given the NRB's significance as a source of water for domestic consumption and ecological health.

**Keywords:** ecologically significant streams; stream discharge trends; hydroclimatic analysis; hydrological indicators; streamflow variability

## 1. Introduction

Access to fresh water resources plays a critical role in sustainable development, particularly in water-scarce regions. Semi-arid areas such as south Texas are highly water-stressed due to limited annual precipitation, high evapotranspirative losses, and prolonged droughts [1,2] as well as competing water demands [3]. Annual rainfall in this region varies from about 20 inches to 40 inches but annual potential evapotranspiration exceeds precipitation by 2 to 5 times [1].

Future climate change effects are not expected to bring any respite to water woes in the region either. The authors of [4] projected a trend towards more dry days as well as warming temperatures in this region, likely leading to increased evapotranspirative losses. Conclusions drawn by the researchers of [5] indicated that there may be no significant change in the amount of precipitation over the next 100 years, but rather that the duration between successive precipitation events will increase in the region.

Several recent studies, for example [6–8], have shown that increasing temperatures may lead to greater unpredictability in precipitation patterns, which will in turn impact streamflow. These studies also emphasized the increased propensity for extreme hydrologic events such as droughts and floods in the future, resulting in negative impacts on water resource management and on aquatic ecosystems, particularly with regards to environmental flows.

Furthermore, the authors of [4] reported that variations in streamflow in the United States are predominantly caused by precipitation. Elsewhere, as in southwestern Australia, streams changed from perennial to ephemeral flow because of sharp declines in precipitation during the mid to late 2000s [9]. Several studies, for instance [10,11] have used the concept of elasticity developed by [12] to evaluate the changes in streamflow in response to changes in hydrologic variables such as precipitation or environmental variables such as land use and land cover.

As such, the south Texas region is characterized by large inter-annual variability in precipitation which results in uncertain streamflows, rendering them an unreliable source of water supply [1]. For instance, highly irregular rainfall characteristics in the NRB result in brief periods of high flows interspersed between long spells of low or zero flows [13]. Approximately 2% of the south Texas region consists of perennial water bodies; despite the sparse occurrence of surface water and the sporadic nature of streams in the region, most urban areas (with the exception of San Antonio) rely on surface water supplies to meet their needs.

Upon the passing of Senate Bill 1 (1997), water planning in Texas became a regional process, which required the Texas Water Development Board [TWDB] to identify 16 regional water planning regions (RWPAs). Of these, the majority of the NRB covers Region L (South Central Texas), but also extends into Region J (Plateau), Region M (Rio Grande), and Region N (Coastal Bend) as shown in Figure 1. The European Union has also adopted a similar style of regional water planning with their River Basin Management Plan [14]. The NRB is also a significant source of water for Region N, particularly for the City of Corpus Christi. According to [15], two of the three reservoirs that serve this city, Choke Canyon Reservoir and Lake Corpus Christi [jointly referred to as CCR/LCC], are located on the NRB. As such, the State of Texas owns all surface water in the state and appropriation thereof is governed by water rights; for example, water stored in the CCR/LCC represents nearly 98% of the water rights in the basin. However, the reliability of these rights is contingent on hydrologic conditions, particularly the occurrence of droughts. Another study [16], found that while droughts are common in this region, their severity has progressively increased; meanwhile, annual inflows to the CCR/LCC system have decreased by 40% from the 1950s to the 1990s. An 83% increase in total water use in Region N over the next 50 years has also been projected, underlining the need for prudent planning of water supply and use.

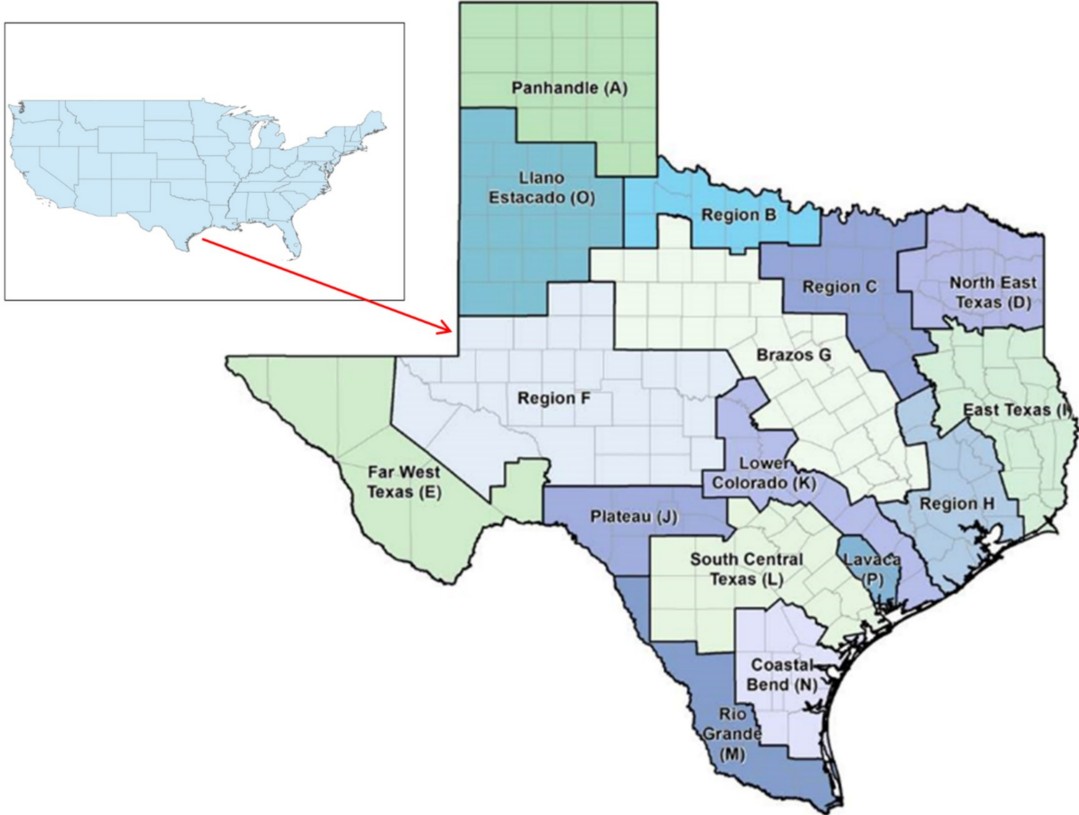

**Figure 1.** Texas location in the United States and Regional Water Planning Areas in Texas [17].

The NRB is unique from a hydrologic perspective due to the complex interactions that occur between streamflows and the subsurface environment, which can significantly alter groundwater recharge and thus indirectly impact water availability. One of the most productive karst aquifer systems in the world, the Edwards Aquifer of Texas, serves the water needs of the south-central Texas region, including the city of San Antonio, and supports several endangered species such as the Texas blind salamander (*Typhlomolge rathbuni*). This system receives significant recharge contributions from above-ground sources such as the Nueces River [18–20]. About 60% of the drainage basin recharges the groundwater system with about 14 billion ft$^3$ of water per year [21]. Roughly 85% of Edwards Aquifer's recharge comes from the watershed while the remainder originates from direct precipitation and subsurface flows from neighboring aquifers such as the Trinity [22]. The Nueces River, in turn, receives contributions from surface runoff as a result of rainfall events as well as groundwater (in the form of springs). The authors of [13] emphasized that the upper portion of the NRB that overlies the Edwards Aquifer outcrop region is "of particular interest to water managers in south-central Texas because appreciable streamflow gains and losses are observed along various reaches in the area". In addition to its hydrologic value to the Edwards Aquifer and the reservoir systems that supply the Coastal Bend region, parts of the Nueces River have also been deemed to have significant ecological value, which is determined by criteria such as biological function, hydrologic function, the presence of threatened, endangered, or unique communities [23].

While there is an abundance of literature on (a) short-term or prevailing hydrologic conditions and water quality in the basin in the form of United States Geologic Survey [USGS] scientific reports (e.g., [13,24,25]), (b) the environmental conditions of downstream sections of the basin such as the Nueces estuary and associated ecosystem (e.g., [26–28]) and (c) groundwater-river water interactions in the basin (e.g., [18,29,30]), few studies have focused on hydrologic trends and their impacts on the basin, particularly on the upstream-most segments designated as ecologically-significant. Additionally,

literature on the impact of variations, if any, in precipitation and land use and land cover (LULC) on streamflow in this basin is lacking.

It is apparent from the preceding discussion that alterations in streamflow of the upper Nueces River and those of its associated tributaries such as the Frio and Sabinal can have adverse consequences on inflows in to the CCR/LCC systems. This in turn will affect water availability in the Coastal Bend region, critically-impact recharge to the Edwards aquifer, as well as detract from the ecological uniqueness of areas designated as such. The sporadic nature of the river as well as its history of recurrent droughts led the authors of [1] to label the climate of this region as 'problematic'. The over-arching goal of this study is, therefore, to evaluate the variability in streamflow in the upper-most part of this basin [near the headwaters] both from a water-planning perspective as well as for stream health assessment reasons. Specifically, we investigate historical spatio-temporal trends in streamflow using a suite of hydrologic indicators to identify any congruity between these metrics and variables such as precipitation and evapotranspiration (ET) in streamflow. The period of 1970 to 2014 was chosen for this study as consistent and reliable data for both streamflow and climate variables are available for this timeframe. Gauges that are part of the USGS HCDN 2009 were used as they reflect on sites with minimal or no human interference or disturbance.

## 2. Methodology

### 2.1. Background Information

A brief review of existing literature is presented in this section. The researchers of [21] assessed the strength of correlation between precipitation and climate indices such as the Pacific Decadal Oscillation [PDO] and El Nino Southern Oscillation [ENSO] in the Nueces, San Antonio, and Guadalupe River basins of south Texas; additionally, they also evaluated statistical trends in precipitation and streamflow between 1950 and 2009. They concluded that while precipitation and streamflow generally showed increasing trends during the period of interest, the correlation between these two variables was poor. Their inferences about hydrologic trends in the NRB are made from data recorded at a single USGS gauge located just upstream of the CCR/LCC system and do not reflect the most recent decades when significant warming was noted to have occurred in this region [6]. The authors of [31] simulated past (1986 to 2005) hydrologic conditions in the basin using a Variable Infiltration Capacity (VIC) model coupled with outputs from a suite of General Circulation Models [GCMs]. The VIC model was calibrated using streamflow data from two USGS HCDN gauging sites in the basin and then run with historical climate data output from four GCMs under the Representative Concentration Pathway [RCP] 4.5 over the period of 1966 to 2005. They showed that the performance of GCMs in capturing observed trends varied spatially but the GCMs were generally in agreement about the reduction in water availability at the outlet of the basin. Earlier studies by the authors of [32,33] suggested declining trends in runoff per unit rainfall in the Atascosa River Basin between 1935 and 1994. The Atascosa River is a tributary of the Frio River, which itself is a tributary of the Nueces River. Recently, the researchers of [7] conducted an extensive study of trends in hydrological drought indicators such as annual 7-day low flows [q7] and annual number of hydrological dry days across major US water resource regions; these areas are designated as USGS Hydrologic Unit Code Level-1 (HUC1) or 2-digit watersheds that encompass major river basins and associated watersheds. They used streamflow data from the USGS HCDN 2009 network [34] which consists of only those gauges that are minimally [or not] affected by human activities or influences. Their results indicated decreasing trends in q7 flows in 'southeastern Texas' but it must be noted that this is reflective on general trends across many HCDN sites in Texas and not confined to the NRB.

### 2.2. Site Description

The NRB study area, shown in Figure 2, is approximately 17,000 square miles in area and spans 20 south Texas counties delimited by Edwards County to the north, Webb County to the south, Maverick

County to the west, and Atascoca County to the east [21]. The watershed is approximately 1600 ft. above mean sea level [35], and the Nueces River flows roughly 315 miles. to the Gulf of Mexico via Nueces Bay [31]. The NCDC classifies Texas into 10 climate divisions based on similarities in vegetation, weather, and climate. Nueces Basin's main areas fall under climate divisions 6 - Edwards Plateau and 9 - South Texas Plains [36]. The region's climate ranges from subtropical steppe to subtropical subhumid and is characterized by semi-arid to arid conditions with hot summers and dry winters [37]. Precipitation in the NRB region averages 20 inches to 30 inches annually [6].

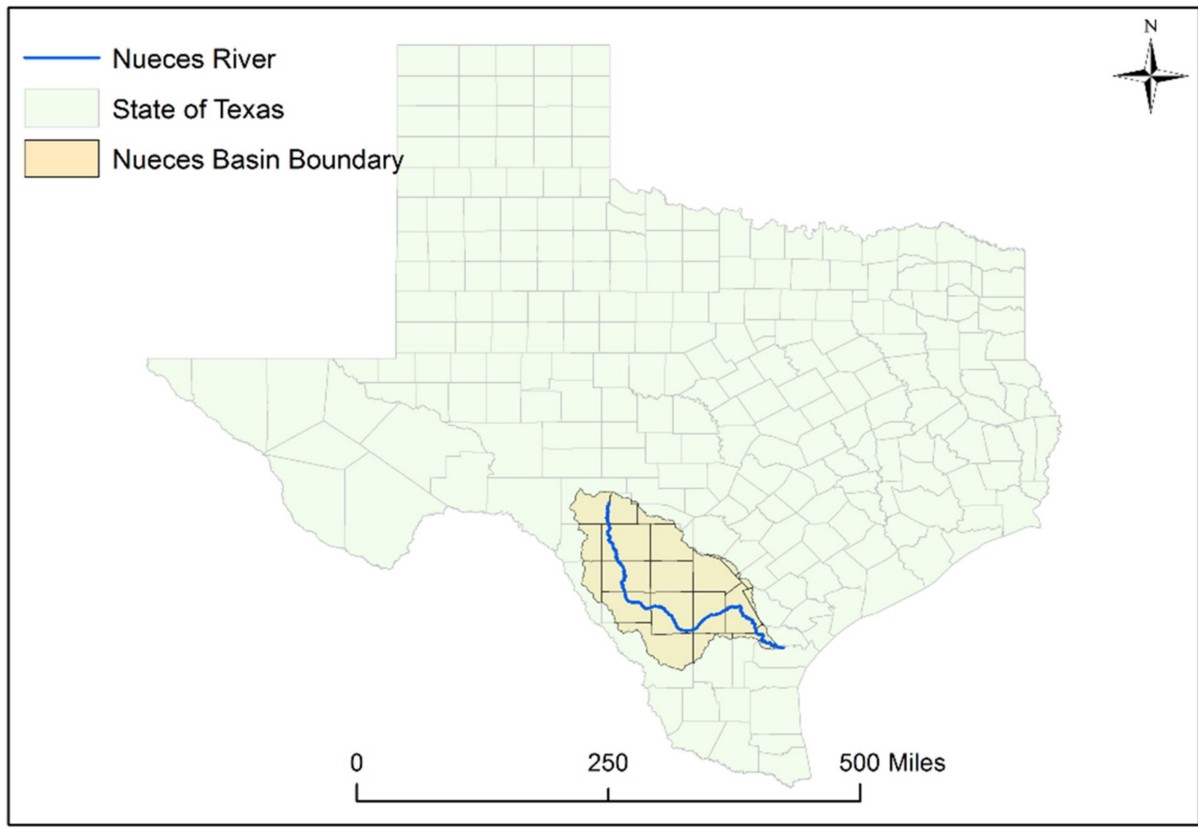

**Figure 2.** Study area: Nueces River Basin in TX, USA.

As of 2009, the year the USGS reevaluated then reported the most recent list of unimpaired gauges in the HCDN [34], the NRB has 36 USGS-managed streamflow gauge sites [38] which continuously gather discharge data. Furthermore, the northern section of the NRB, which as previously mentioned is primarily covered by Region L, has 31 streams that are designated as ecologically significant [23]. As indicated in Figure 3, four of the six gauges that were used in the study are located on ecologically unique stream segments while all of the chosen gauge sites contribute to Edwards Aquifer recharge. Beyond the hydrologic function of the streams that is depicted in Figure 3, Table 1 further highlights the numerous invaluable roles the stream segments fulfill based on the TWDB's ecologically-significant designation criteria [39].

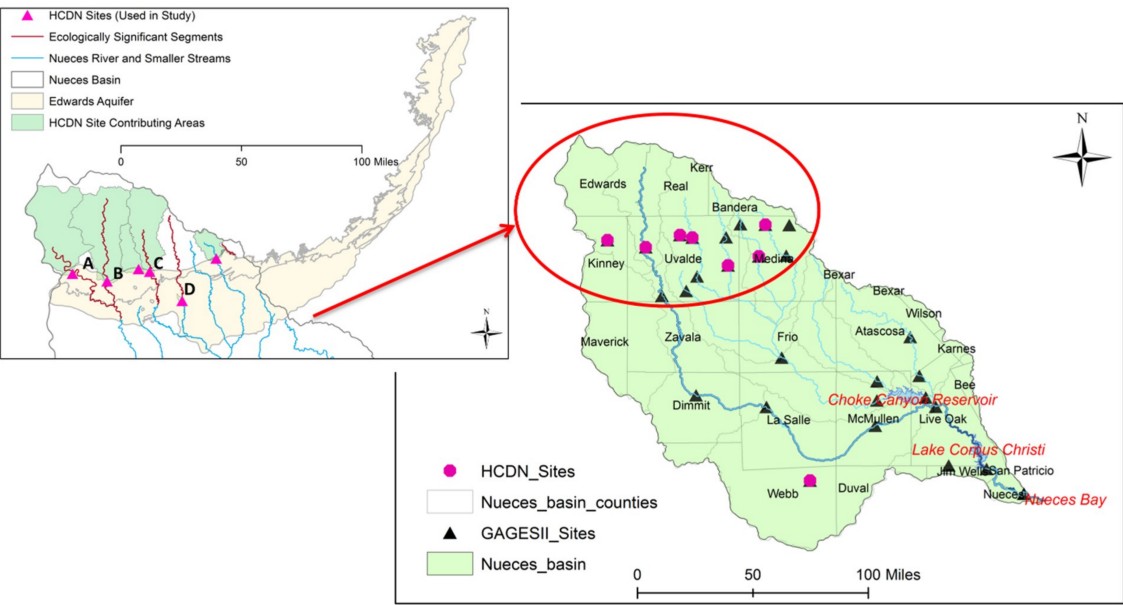

**Figure 3.** Hydroclimatic Data Network gauge locations (see Table 1 for gauge information).

**Table 1.** Functions of the significant streams in the study area.

| Identifier | Gauge ID | Stream | Ecologically Significant Streams Criteria | | | | |
|:---:|:---:|:---:|:---:|:---:|:---:|:---:|:---:|
| | | | BF | HS | RC | HQ | TS |
| A | 08190500 | West Nueces River | | ✓ | | | ✓ |
| B | 08190000 | Nueces River | ✓ | ✓ | | ✓ | |
| C | 08195000 | Frio River | ✓ | ✓ | ✓ | ✓ | |
| D | 08198500 | Sabinal River | ✓ | ✓ | | ✓ | |

[1] BC – Biological function; HC – Hydrologic function; RC – Riparian conservation areas; HQ – High water quality, exceptional aquatic life, and high aesthetic value; TS – Threatened or endangered species, unique communities.

## 2.3. Data Sources

The period 1970 to 2014 was used to determine the hydroclimatic variability for streamflow and precipitation in the study area. That particular timeframe was chosen because (a) it provided consistent and reliable data for both hydrological and climate variables, (b) the 45-year duration ensured statistical legitimacy for the study since authors of [40,41] indicated that at least a 15- to 25-year period is required to determine statistical validity for streamflow trend analysis, and (c) we aimed to extend the most recent time series for hydroclimatic analyses in the region done by [21] which went until the year 2009, the year the USGS made the HCDN revisions.

United States Geological Survey-verified continuous daily mean discharge data were acquired from the HCDN online database [42] in units of cubic feet per second (cfs). Using HCDN gauges ensured that minimal anthropogenic influence occurred thereby increasing the likelihood of attributing any streamflow variation to factors other than direct human influence. The work done by [43] documented a comprehensive USGS-sanctioned Geographic Information Systems (GIS) analysis of U.S. land-use trends from 1974 through 2012 as shown in Figure 4. The findings showed minimal land modification from under conservation to developed production in the northern NRB during the aforementioned time period, thereby fulfilling the main HCDN criteria. The northern NRB was also delineated into the subbasins that contribute to each stream gauge, also shown in Figure 4, to focus on specific areas of LULC change in relation to the locations of the stream gauges to estimate any likely effects on the HCDN sites if trends were later determined. The most notable land alterations, from conservation to developed, were relatively small and occurred during the 1974 to 1982 time period in

areas that contribute to the 08196000 and 08198500 gauges. Additionally, the range of daily mean flows measured at each gauge over the 45-year study period is shown in Figure 5. Flow variations have been presented gauge-wise, from westernmost (gauge 08195000; left corner) to easternmost (gauge 08200000; right corner) in this figure. These violin plots are essentially a combination of box plots and kernel density plots that allow for visualization of the shape of the distribution and identification of possible mass clusters. The median of daily flows over the 45-year period have been shown using the diamond-plus symbol; the median flows for all of the selected gauges are less than 150 cfs. It can also be seen from Figure 5 that a large percent of the mass is concentrated on the low flows with gauge 08196000 showing the most pronounced 'bulge' at the low-flow ends. The large variability in flows, spanning four orders of magnitude is also evident from the figure.

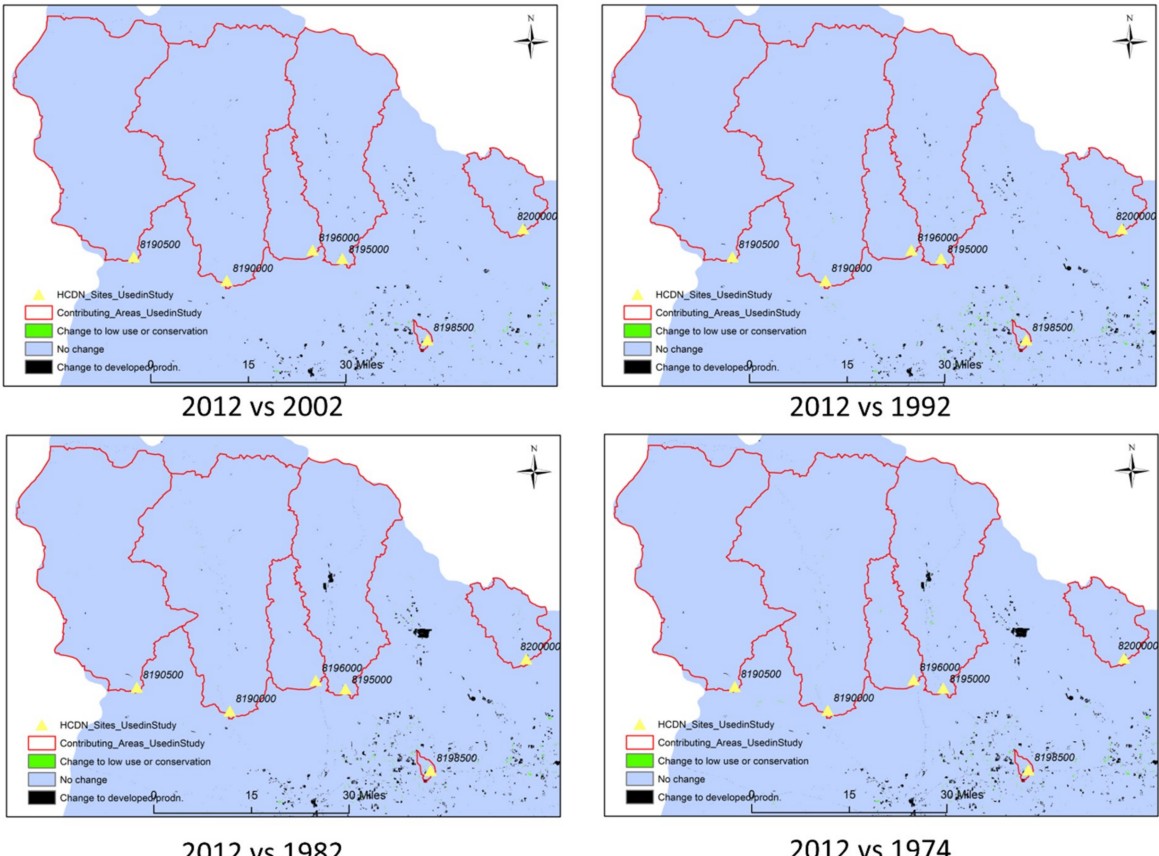

**Figure 4.** Land use changes from 1974 to 2012.

Daily precipitation depths, measured in inches, were acquired from National Oceanic and Atmospheric Administration's (NOAA) NCDC online database [44]. The choice of precipitation gauge stations used in the study was made entirely on the basis of availability of serially-continuous data. Some stations had near-complete records for the 45-year period of analysis but there were occasional gaps in the series. These gaps were filled in using spatial interpolation of synchronous observations from adjacent stations. Other stations did not have appreciable completeness or continuity, so data from multiple stations had to be consolidated in these cases. Therefore, the rationale here is the completeness of the precipitation data (without gaps) rather than the nearness of precipitation station to the discharge measurement location. For instance, Uvalde and Medina Counties had sufficient precipitation data while Kinney County did not, so gauges from nearby Zavala County was used as its substitute, as shown in Table 2.

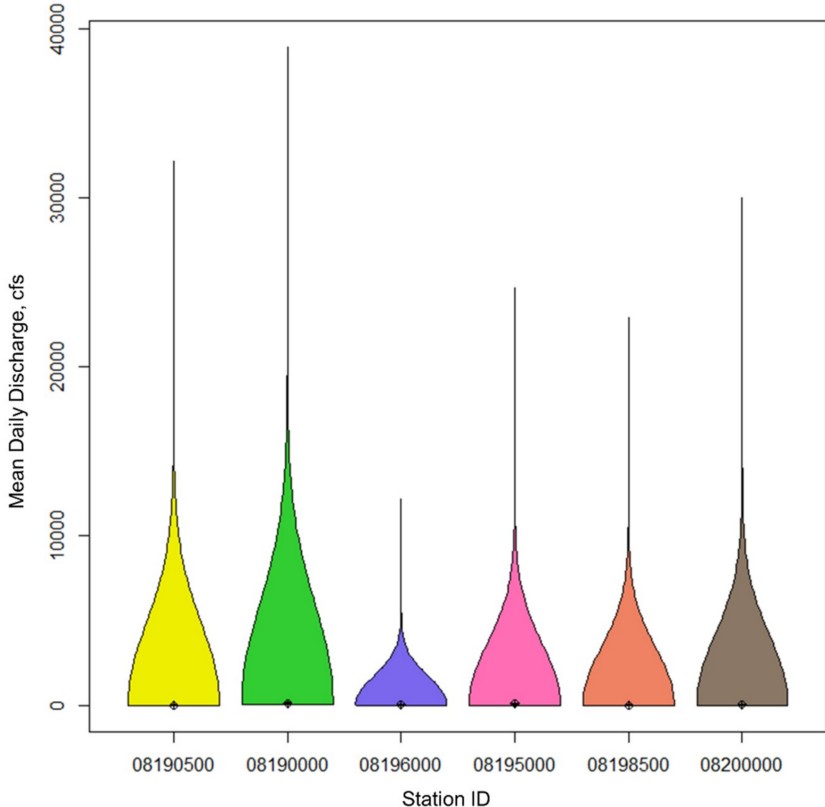

**Figure 5.** Streamflow variability from 1970–2014 (median values for this time period are shown using the diamond plus symbol).

**Table 2.** Consolidated precipitation stations.

| County | Substituted for | Station Name | Station ID | Latitude | Longitude | Corresponding Gauge |
|--------|-----------------|--------------|------------|----------|-----------|---------------------|
| Zavala | Kinney | La Pryor TX US | GHCND:USC00414920 | 28.950000 | −99.833330 | 08190500 |
| | | Crystal City TX US | GHCND:USC00412160 | 28.683330 | −99.833330 | |
| | | Chaparrosa Ranch TX US | GHCND:US1TXZV0001 | 28.885800 | −99.996900 | |
| | | McNally TX US | GHCND:US1TXZV0008 | 28.839700 | −99.924200 | |
| Uvalde | N/A | Sabinal TX US | GHCND:USC00417873 | 29.333330 | −99.483330 | 08190000 |
| | | Uvalde Tx US | GHCND:USC00419265 | 29.216670 | −99.766670 | 08195000 |
| | | Utopia Tx US | GHCND:USC00419260 | 29.616670 | −99.516670 | 08196000 |
| | | Uvalde 3 SW Tx US | GHCND:USC00419268 | 29.216670 | −99.750000 | 08198500 |
| Medina | N/A | Riomedina TX US | GHCND:USC00417628 | 29.466670 | −98.866670 | 08200000 |
| | | Lytle 3 W US | GHCND:USC00415454 | 29.233330 | −98.800000 | |
| | | Hondo TX US | GHCND:USC00414254 | 29.336500 | −99.138600 | |
| | | Natalia TX US | GHCND:USC00416205 | 29.200000 | −98.866670 | |

Evapotranspiration is a major hydrological output in the water budget of south Texas where annual PET can exceed annual precipitation by 2 to 5 times [1]. Evapotranspiration data were adapted from [45] and evaluated for each selected gauge location. The author of [45] estimated the average change in actual ET trends over the contiguous United States for the period 1979 to 2015 using complementary relationship (CR) based ET estimation methods suggested by [46]. The research in [45] validated the ET estimates using precipitation and runoff data from the Parameter-Elevation Regressions on Independent Slopes Model (PRISM) and USGS Hydrologic Unit Code Level-6 (HUC6) or 12-digit watersheds respectively.

## 2.4. Hydroclimatic Data Analysis

To evaluate the relationships between the hydroclimatic variables, a range of hydrological assessment metrics specific for nonparametric datasets where applicable were used: modified Mann-Kendall test (MMK), annual minimum and 7-day low flow indicators, streamflow elasticity, and drought indices [Streamflow Drought Index (SDI) and Standardized Precipitation Index (SPI)]. Using a number of indicators was critical to determine potential relationships between hydroclimatic variables which could ultimately help indicate the extent to which variations in precipitation affected stream discharge. Consequently, trend analysis using MMK as well as hydrological sensitivity assessments between precipitation and streamflow were investigated along with streamflow elasticity and drought indices to understand the possible relationships. Low flows, which are necessary for maintaining certain essential ecological functions and overall stream health, were assessed using the previously stated low flow metrics and analyzed in conjunction with the trends and hydrological sensitivity to determine if potential streamflow variations were mainly associated with changes in precipitation.

### 2.4.1. Trend Analysis (MMK)

The MMK is a reliable and popular nonparametric trend analysis tool that was used to detect the hydroclimatic trends. Numerous researchers, for example [21,47–53], have used various forms of the Mann-Kendall test. The authors of [50] as well as [51] indicated that for accurate hydroclimatic trend analyses the input data must be serially independent to prevent serial correlation or autocorrelation. Serial correlation occurs when errors in a particular timespan transfer into subsequent time periods [54], which could cause an overestimation or underestimation of trends [50,51]. Therefore, the MMK, which was derived by [55] and designed to account for serial correlation by modifying the variance of the original Mann-Kendall test [51], was chosen.

The original Mann-Kendall test statistic, which is shown in Equation 1, was developed by [56,57]. A detailed explanation along with the appropriate equations for the original trend analysis method Equations (1) and (2) where $n$, $X_i$, $X_j$, and $D$ are sample size, consecutive data values, and the difference between successive values respectively were presented by authors of [10,47,51].

$$S = \sum_{i=1}^{n-1} \sum_{j=i+1}^{n} sgn(X_j - X_i) \tag{1}$$

$$sgn(X_j - X_i) = \begin{cases} 1 \ if \ D > 0 \\ 0 \ if \ D = 0 \\ -1 \ if \ D < 0 \end{cases} \tag{2}$$

The aforementioned authors, [10,47,51], also provided the test statistic Equation (3) and equations for the modified version of the Mann-Kendall test as shown in Equation (3) through Equation (7) where $V(S)^*$ is the modified variance from $V(S)$ in Equation (5) and $r_k$ is the lag-k autocorrelation coefficient Equation (7):

$$Z = \begin{cases} \frac{S-1}{\sqrt{V(S)*}} \ for \ S > 0 \\ 0 \ for \ S = 0 \\ \frac{S+1}{\sqrt{V(S)*}} \ for \ S < 0 \end{cases} \tag{3}$$

$$V(S)* = V(S)\frac{n}{n*} \tag{4}$$

$$V(S) = \frac{n(n-1)(2n+S)}{18} \tag{5}$$

$$\frac{n}{n*} = 1 + \frac{2}{n(n-1)(n-2)} \sum_{i=1}^{n-1} (n-i)(n-i-1)(n-i-2)r_i \tag{6}$$

$$r_k = \frac{\frac{1}{n-k} \sum_{i=1}^{n-k} \left(X_i - \overline{X}\right)\left(X_{i+k} - \overline{X}\right)}{\frac{1}{n} \sum_{i=1}^{n} \left(X_i - \overline{X}\right)^2} \tag{7}$$

The null hypothesis ($H_0$) for the MMK is "no monotonic trends exist in the time series" [10], and it was rejected at a $\leq 0.01$ significance level, while the alternate hypothesis ($H_A$) is "there are monotonic trends in the time series". The hydroclimatic trends were determined using a combination of their Z values and corresponding *P* values. Increasing and decreasing trends were represented by positive and negative Z values respectively [10,47,48].

Hydrological trend analysis was performed for minimum, median, and maximum stream discharge and precipitation over monthly, seasonal, and annual temporal scales. Annual assessments were done on a calendar year scale. June to August represented summer months, September to November was fall, December to February signified winter, and March to May denoted spring. Trend assessments were done for the periods of 1970 to 2014. Additionally, we performed trend analyses for the period 1994 to 2014. Trends in mean annual temperatures in South Texas between 1930 and 2001 were studied by [1]. These authors fit regression equations for temperature based on data collected from 16 stations across the study region, which generally overlaps the NRB. Their findings suggest an upward trend in temperature around the mid-1990s, thus serving as the rationale behind distinguishing the 1994 to 2014 period for separate analysis. The results from both sets of trend analysis were later compared against each other. The "fume" package in R [58] was used to perform the MMK.

### 2.4.2. Low Flow Metrics (Annual Minimum and 7-day Low Flows)

The annual 7-day low flow of a stream is the lowest average flow that occurs within a consecutive 7-day period during any given year while the annual minimum flow measures the lowest discharge for each year. Low flow plays an essential role in the ecological balance of riverine systems. Such flow is critical for maintaining environmental flows, water quality, biodiversity, natural migration trends, ecosystem integrity, and overall stream health. The south Texas region is known to experience harsh summer conditions, unpredictable precipitation patterns, and widely variable flow, as exemplified in Figure 6, where representative examples were chosen. Low flow assessments for the selected gauges in the NRB were therefore necessary because of the important hydrological, ecological, and various other functions the stream segments provide to the downstream regions due to their ecologically significant designations. The low flow metrics analysis was done using the "FlowScreen" package in R, which was developed by [59], while the MMK was used to determine the trends from the low flow indicators.

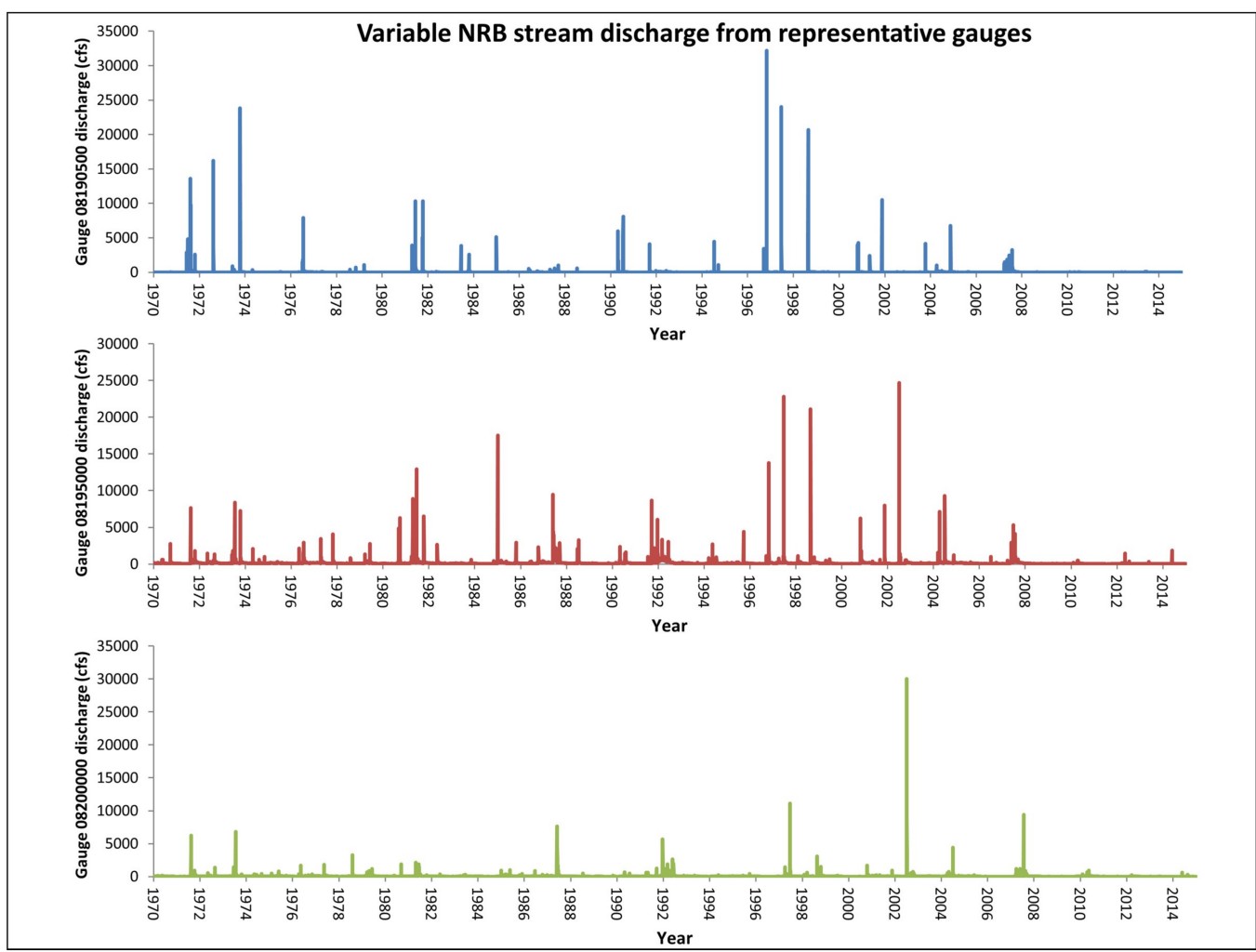

**Figure 6.** Streamflow from the most western (top), central (middle), and eastern (bottom) gauges in the NRB.

### 2.4.3. Streamflow-Precipitation Elasticity

Streamflow elasticity is a measure of mean streamflow variation in proportion to changes in another driving variable such as precipitation [11,60]. The analysis was used because it quantifies stream discharge sensitivity to environmental changes [11,60]. The author of [12] first presented the streamflow elasticity concept for precipitation variations using the expression shown in Equation (8), where $Q$ and P are stream discharge and precipitation respectively [11]. The researchers of [60] refined the concept to standardize results using median as well as mean streamflow and precipitation values as shown in Equation (9). In Equation (9), $\overline{P}$, $\overline{Q}$, $Q_t$, and $P_t$ denote mean annual streamflow and precipitation as well as yearly stream discharge and precipitation respectively [10,11,60]. To perform the elasticity calculations, the stream discharge and precipitation units were both standardized to depths in inches.

$$\epsilon_p(P,\ Q) = \frac{dQ/Q}{dP/P} = \frac{dQ}{dP}\frac{P}{Q} \tag{8}$$

$$\epsilon_p = median\left(\frac{Q_t - \overline{Q}}{P_t - \overline{P}}\frac{\overline{Q}}{\overline{P}}\right) \tag{9}$$

### 2.4.4. Drought Indices (SDI and SPI)

The SDI and SPI were evaluated to assess the NRB's hydrologic extremes. The developers of the SDI, [61], along with [62], noted that the indices require minimal amounts of data and computational effort, which make them highly advantageous over earlier indices developed for determining water availability and variability patterns. The SDI creators, [61], indicated that to compute the SPI a times series of monthly precipitation, which is denoted by $P_{i,j}$ where $i$ is the hydrological year and $j$ is the month in the hydrological year, is required to produce the series as shown in Equation 10. From the series depicted by Equation (10), a sequence where $R_{j,k}$ represents the depth of cumulative precipitation for the $k$-th reference period, which consists of overlapping time intervals of 3, 6, 9, and 12 months in the $i$-th hydrological year, and reference periods $k$ is derived. In the SPI calculation shown in Equation (11), the mean and standard deviation of the cumulative precipitation depths are represented by $\overline{R}_k$ and $S_{R,k}$ respectively for the $k$-th reference period for statistical estimations over an extensive timespan.

$$R_{i,k} = \sum_{j=1}^{3k} P_{i,j}\ i = 1,2,\ldots,\ j = 1,2,\ldots,12,\ k = 1,\ 2,\ 3,\ 4 \tag{10}$$

$$SPI_{i,k} = \frac{R_{i,k} - \overline{R}_k}{S_{R,k}}\ i = 1,2\ldots,\ k = 1,2,3,4 \tag{11}$$

Additionally, during drought indices computation the usually skewed probability distributions of hydroclimatic data require transformation to a normal distribution, which is typically done using a Gamma distribution [62]. The computational representations for the logarithmic transformation are shown in Equations (12) and (13) where $\ln(R_{i,k})$, $\overline{w}_k$, and $S_{w,k}$ are the natural logarithms of cumulative precipitation, mean, and standard deviation respectively over a lengthy timeframe.

$$SPI_{i,k} = \frac{w_{i,k} - \overline{w}_k}{S_{w,k}}\ i = 1,2\ldots,\ k = 1,2,3,4 \tag{12}$$

$$w_{i,k} = \ln(R_{i,k})\ i = 1,2,\ldots,\ k = 1,\ 2,\ 3,\ 4 \tag{13}$$

Converting the time series to normal distributions also ensures that they have a mean of zero and standard deviation of 1, which allows for easy interpretation of the results. Index values between $-1$ and $+1$ represent average conditions, $<-1$ indicates varying degrees of drought, and $>+1$ shows wetter than average conditions. Furthermore, the SPI and SDI computations use the same methodology [61] but different univariate hydroclimatic input data. Precipitation and streamflow data are required

for the SPI and SDI respectively. When both indices are analyzed together, the SDI lags the SPI. The drought indices were calculated using a 12-month time-scale as droughts usually take multiple seasons or even years to manifest in hydrologic variables.

## 3. Results

The results for each hydrological assessment metric are presented sequentially, from the westernmost gauge, 08190500, to easternmost gauge, 08200000, in Kinney and Medina Counties respectively. Additionally, the notations $Q_{min}$, $Q_{med}$, and $Q_{max}$ correspondingly represent minimum, median, and maximum discharge.

*3.1. MMK Trend Analyses – 1970 to 2014*

Few statistically significant ($p \leq 0.01$) precipitation trends were detected for the entire study period. Significant decreases in annual maximum precipitation were only identified in Uvalde County during the spring and fall seasons as well as the months of April. Contrastingly, many statistically significant annual and seasonal stream discharge trends were found for streamflow, as indicated in Table 3. An overwhelmingly greater number of decreasing trends were identified in the $Q_{min}$ and $Q_{med}$ flows when compared to $Q_{max}$. Decreasing flows were most often detected in $Q_{min}$ and $Q_{med}$ in the NRB's central to easternmost gauges (08196000, 08195000, 08198500, and 08200000) found in Uvalde and Medina Counties for various temporal periods. Except for a few instances, the previously listed gauges simultaneously experienced decreasing trends over the annual time scale, fall and winter periods, as well as most of the months that corresponded to the said seasons (September, October, November, December, and January) for both $Q_{min}$ and $Q_{med}$. The gauges that are located in the western part of the basin (08190500 and 08190000) collectively experienced the least number of statistically significant trends for any flow level. No trends were detected in the westernmost gauge, 08190500, over any of the temporal scales during the 1970 to 2014 timespan, while very few trends were identified for gauge 08190000 throughout the same period. Statistically significant decreases in $Q_{max}$ were only found in gauges 08195000 (winter, fall, January, and October) as well as 08200000 (August, September, October, and November). July was the only month in which no statistically significant trends were identified.

**Table 3.** Annual and seasonal NRB streamflow trends from 1970 to 2014 ([NT] no significant trend; [↓] statistically significant decreasing trend).

| | $Q_{min}$ | | | | |
| --- | --- | --- | --- | --- | --- |
| **Gauge ID** | **Annual** | **Spring** | **Summer** | **Fall** | **Winter** |
| 08190500 | NT | NT | NT | NT | NT |
| 08190000 | NT | ↓ | NT | ↓ | NT |
| 08196000 | ↓ | NT | ↓ | ↓ | ↓ |
| 08195000 | ↓ | ↓ | NT | ↓ | ↓ |
| 08198500 | ↓ | NT | NT | ↓ | ↓ |
| 08200000 | ↓ | NT | NT | ↓ | ↓ |
| | $Q_{med}$ | | | | |
| **Gauge ID** | **Annual** | **Spring** | **Summer** | **Fall** | **Winter** |
| 08190500 | NT | NT | NT | NT | NT |
| 08190000 | ↓ | NT | ↓ | NT | NT |
| 08196000 | ↓ | ↓ | NT | ↓ | ↓ |
| 08195000 | ↓ | ↓ | NT | ↓ | ↓ |
| 08198500 | ↓ | NT | NT | NT | ↓ |
| 08200000 | NT | NT | ↓ | ↓ | ↓ |

**Table 3.** *Cont.*

| | $Q_{max}$ | | | | |
|---|---|---|---|---|---|
| Gauge ID | Annual | Spring | Summer | Fall | Winter |
| 08190500 | NT | NT | NT | NT | NT |
| 08190000 | NT | NT | NT | NT | NT |
| 08196000 | NT | NT | NT | NT | NT |
| 08195000 | NT | NT | NT | ↓ | ↓ |
| 08198500 | NT | NT | NT | NT | NT |
| 08200000 | NT | NT | NT | NT | NT |

*3.2. MMK Trend Analyses – 1994 to 2014*

No statistically significant precipitation trends were found for the last 20 years of the study, but significant decreasing streamflow trends were detected for annual and seasonal periods, as shown in Table 4. The 1994 to 2014 period had more significant declines in $Q_{max}$ in comparison to the full 45-year assessment. Statistically significant decreases in $Q_{min}$, $Q_{med}$, and $Q_{max}$ were more often detected in the westernmost to central gauges, which again differed from the 1970 to 2014 timespan, where the central to easternmost gauges largely experienced a greater number of declines. Decreases in $Q_{med}$ and $Q_{max}$ were also identified in each gauge except those on the eastern end of the basin in the fall and winter seasons, while only gauges 08196000 and 08200000 did not have significant declines during winter in $Q_{min}$. Few statistically significant trends were found for the annual stream discharge (08196000, 08195000, and 08200000 for $Q_{med}$ as well as 08190500 and 081900000 for $Q_{max}$), the spring seasons (08190500, 08195000, and 08198500 in $Q_{min}$), and summer periods (08200000 in $Q_{min}$ only). The months that collectively experienced the most streamflow declines in $Q_{min}$, $Q_{med}$, and $Q_{max}$ were October, November, and December. However, the months of April to September which practically covered the entire spring and summer seasons together had the least decreases in streamflow in $Q_{min}$, $Q_{med}$, and $Q_{max}$. No statistically significant trends were found for the months of May and August.

**Table 4.** Annual and seasonal NRB streamflow trends from 1994 to 2014 ([NT] no significant trend; [↓] statistically significant decreasing trend).

| Gauge ID | $Q_{min}$ | | | | |
|---|---|---|---|---|---|
| | Annual | Spring | Summer | Fall | Winter |
| 08190500 | NT | ↓ | NT | NT | ↓ |
| 08190000 | NT | NT | NT | NT | ↓ |
| 08196000 | NT | NT | NT | NT | NT |
| 08195000 | NT | ↓ | NT | ↓ | ↓ |
| 08198500 | NT | ↓ | NT | NT | ↓ |
| 08200000 | NT | NT | ↓ | NT | NT |
| **Gauge ID** | **$Q_{med}$** | | | | |
| | Annual | Spring | Summer | Fall | Winter |
| 08190500 | NT | NT | NT | ↓ | ↓ |
| 08190000 | NT | NT | NT | ↓ | ↓ |
| 08196000 | ↓ | NT | NT | ↓ | ↓ |
| 08195000 | ↓ | NT | NT | ↓ | ↓ |
| 08198500 | NT | NT | NT | NT | ↓ |
| 08200000 | ↓ | NT | NT | NT | NT |

**Table 4.** *Cont.*

| Gauge ID | Q$_{max}$ | | | | |
|---|---|---|---|---|---|
| | Annual | Spring | Summer | Fall | Winter |
| 08190500 | ↓ | NT | NT | ↓ | ↓ |
| 08190000 | ↓ | NT | NT | ↓ | ↓ |
| 08196000 | NT | NT | NT | ↓ | ↓ |
| 08195000 | NT | NT | NT | ↓ | ↓ |
| 08198500 | NT | NT | NT | NT | ↓ |
| 08200000 | NT | NT | NT | NT | NT |

### 3.3. Low Flow Indicators

Statistically significant decreasing trends were detected in both the annual minimum and 7-day low flows for two central gauges (08196000 and 08195000) and the easternmost gauge (08200000). Contrastingly, no significant trends were found for the westernmost gauges (08190500 and 08190000) as well as one central gauge (08198500) for the low flow metrics analyses.

### 3.4. Streamflow-Precipitation Elasticity

Table 5 shows the streamflow-precipitation elasticity that was calculated for each gauge site. The elasticity of gauges 08190500 and 08198500 with Zavala and Uvalde Counties respectively were close to 1:1 ratios. In both cases, the hydroclimatic sensitivity essentially indicated that a 1% increase in annual precipitation would have theoretically resulted in an almost equivalent increase in annual stream discharge. For the remaining gauges however, there was less sensitivity between the hydroclimatic variables since the elasticity coefficients were less than 1. The results signaled that annual precipitation fluctuations had a smaller impact on yearly streamflow. In theory, a 1% variation in annual precipitation would have produced a 0.47%, 0.60%, 0.41%, and 0.72% change in annual streamflow for gauges 08190000, 08196000, 08195000, and 08200000 respectively.

**Table 5.** Streamflow precipitation elasticity for each gauge.

| Gauge ID | Gauge Name | Elasticity | Counties |
|---|---|---|---|
| 08190500 | W Nueces Rv nr Brackettville, TX | 0.99 | Zavala |
| 08190000 | Nueces Rv at Laguna, TX | 0.47 | |
| 08196000 | Dry Frio Rv nr Reagan Wells, TX | 0.60 | Uvalde |
| 08195000 | Frio Rv at Concan, TX | 0.41 | |
| 08198500 | Sabinal Rv at Sabinal, TX | 0.95 | |
| 08200000 | Hondo Ck nr Tarpley, TX | 0.72 | Medina |

### 3.5. Drought Indices (SDI and SPI)

The SPI and SDI were evaluated to assess NRB's hydroclimatic variables for the 45-year study period for anomalous hydrologic occurrences such as droughts in the SDI, which is defined as having results that are <−1 [62]. The changes in precipitation were later followed by similar variations in streamflow. In each corresponding stream discharge and precipitation dataset, there were multiple fluctuations between wetter and drier conditions as well as droughts in varying degrees of severity. As shown in Figure 7, normal to wetter conditions dominated in the first 30 years for both indices, with few recorded instances of droughts. When droughts occurred during the first 30 years, they were mild, most often in the SPI, and had short durations. In the last 15 years of each dataset however, downward trends were evident for both drought indices. The SDI and SPI also revealed that each stream segment and its corresponding county underwent multiple periods of abnormally dry conditions during the last 15 years of the 45-year study period. In each case, more severe drought conditions were first manifested in the SPI due to fluctuations in precipitation then lagged by less harsh droughts in the SDI.

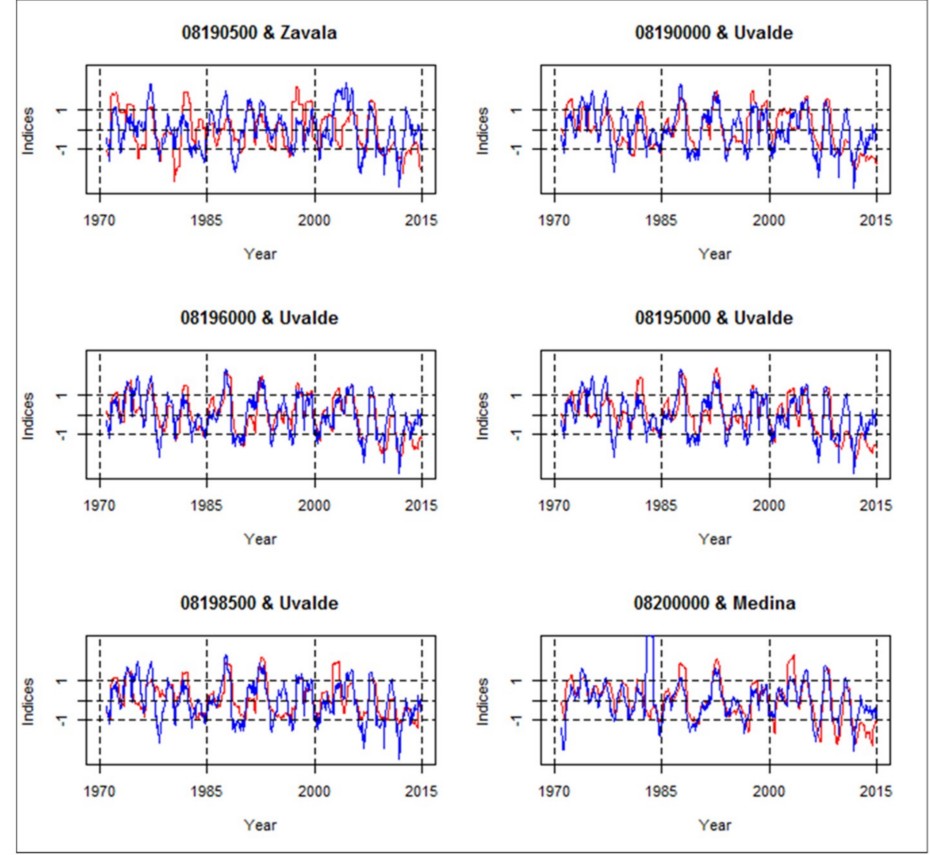

**Figure 7.** Streamflow Drought Index (Red) and Standardized Precipitation Index (Blue).

*3.6. ET Trends*

Table 6 shows the ET results for each gauge location that was extracted from [45], who estimated the average changes in ET over the conterminous United States for the period 1979 to 2015. The effect of the ET changes ranged from positive to negative values, which indicated increases and decreases in ET respectively, across the western to eastern areas of the watershed. The data showed that the westernmost gauges (08190500 and 08190000) were most affected by ET. The central gauge sites (08196000 and 08195000) were practically unaffected by changes in ET while the eastern ones (08198500 and 08200000) were the least impacted.

**Table 6.** Upper NRB change in actual ET from 1979 to 2015 adapted from Szilagyi (2017).

| Gauge ID | Gauge Name | Δ Actual ET (in/year) | Counties |
|---|---|---|---|
| 08190500 | W Nueces Rv nr Brackettville, TX | 0.048 | Zavala |
| 08190000 | Nueces Rv at Laguna, TX | 0.030 | |
| 08196000 | Dry Frio Rv nr Reagan Wells, TX | 0.003 | Uvalde |
| 08195000 | Frio Rv at Concan, TX | 0.001 | |
| 08198500 | Sabinal Rv at Sabinal, TX | −0.005 | |
| 08200000 | Hondo Ck nr Tarpley, TX | −0.024 | Medina |

## 4. Discussion

This study used a collection of hydrologic indicators to robustly investigate the hydroclimatic relationships of ecologically significant streams in the upper regions of the NRB and the potential impacts of hydrologic alterations for the period 1970 to 2014. The novel aspects of this study are that we (a) have focused on the upper-NRB where the ecologically significant segments lie and where no literature regarding the impact of hydrologic changes on these segments exists and (b) use a diverse

suite of hydrological indicators to capture the response and sensitivity of the river to changes in external drivers. We have also updated the most recent time series used in hydroclimatic analyses of the NRB by the authors of [21] from 2009 to 2014. Additionally, we make recommendations to policy-makers and natural resource managers on holistic and sustainable approaches to water resource planning in the region.

The results showed occurrences of statistically significant streamflow declines in the selected gauges, but precipitation in the corresponding counties overwhelmingly showed no trends. The MMK assessments detected significant decreasing stream discharge trends in the central to easternmost gauges, mainly in $Q_{min}$ and $Q_{med}$ for the 1970 to 2014 period, as well as more uniform declines over each flow level ($Q_{min}$, $Q_{med}$, and $Q_{max}$) for the westernmost and central gauges during the most recent 20 years of the study. The decreases in stream discharge were consistent with earlier studies of the spatiotemporal hydroclimatic relationships in the basin in [32,33], which concluded that runoff per unit precipitation in the Atascosa River Basin, a subbasin in the upper NRB, from 1935 and 1994 showed declining trends.

The droughts that were experienced in the NRB, particularly ones during last 15 years, were first manifested in the SPI followed by the SDI. The SPI, a precipitation based index, detected the trends in drought early on; whereas, the SDI, which is based on streamflow, also identified the drought trends but with an expected lag or delay. The lag between the two drought indices suggested that a relationship exists between precipitation and streamflow. The degree to which the hydroclimatic variables are related was shown to be moderate to high by the streamflow-precipitation elasticity. However, because statistically significant decreases in precipitation were only detected in Uvalde County during spring, fall, and the months of April for the 45-year study period, it was clear that precipitation was not the only reason for stream discharge declines. The authors of [63] indicated that precipitation that eventually becomes streamflow generally encounters a lag period because of factors such as basin size and antecedent soil moisture. Consequently, the precipitation reductions in the fall seasons coupled with extended lag times could have led to streamflow declines in the successive fall and winter periods. Plus, with the statistically significant declines in spring precipitation, decreased flows in $Q_{min}$ and $Q_{med}$ were mainly detected in gauges in Uvalde County during that season. Summer and its corresponding months virtually showed very few statistically significant decreasing trends in streamflow for both 45- and 20-year time periods. Texas' semi-arid to arid climate experiences hot summers [37] and variable precipitation [1]. During summer months, many Texas streams suffer from reduced and in some cases zero flows. Therefore, the typical summer conditions could possibly explain why practically no streamflow trends were detected during that time period. However, due to the demonstrable lack of statistically significant precipitation trends, it was apparent that other reasons contributed to the declining streamflows.

The authors of [63] noted that stream discharge trends sometimes do not reflect changes in precipitation because of a combination of climatological variables. The dataset from [45] assessed changes in actual ET trends from 1979 to 2015, and it was used as a surrogate for temperature variation for each gauge location. The effects of ET on streamflow, however, were not conclusive for either the annual 45- or 20-year periods. The assessments from [45] indicated that changes in ET trends were positive in the western areas of the basin, which suggests that ET had a greater effect on the westernmost gauges (08190500 and 08190000). The central to easternmost gauges (08196000, 08195000, 081985000, and 08200000) all experienced little to no impact from ET as the [45] dataset showed negligible changes in ET trends in those areas. The ET results were not indicative of the likely locations for decreasing discharge in the 45-year period as trend analyses showed that streamflow declines overwhelmingly occurred in the central to easternmost gauges. When the changes in ET trends were compared with the annual decreases seen in the last 20 years, only in $Q_{max}$ did statistically significant declines often occur in the western portion of the basin, where changes in ET trends were also positive. In the other flow levels, no significant trends were found for $Q_{min}$ while $Q_{med}$ only experienced decreases in the locations that span the central to eastern gauges. Similar to the 45-year period, the

trend analyses in $Q_{med}$ for the 20-year timespan conflicted with the ET findings which showed that the central to eastern areas had the least ET effect. The general discrepancy between the streamflow and changing ET trends was likely because the ET assessments were not basin-specific as noted by the researchers of [64]. The authors suggested that hydroclimatic variables should be analyzed at the local rather than regional scale for the most accurate analysis since trend effects on extensive scales essentially normalize the final result which may not account for the subtle variances on a localized level. Consequently, other environmental factors were considered.

As previously mentioned, [43] documented U.S. LULC trends from 1974 through 2012 using GIS analysis. Different land uses either impede or facilitate stream discharge. Therefore, investigating temporal changes in land use from being under conservation to being developed with time was necessary to indicate if the NRB region was affected by manmade influences that would have altered the basin's hydrological characteristics, hence affecting the area's HCDN status. The analysis indicated that the NRB had undergone LULC alterations, particularly in its southern end, as shown in Figure 4. More importantly however, the analysis revealed that the upper-most basin regions, which contribute to the gauges in our study, remained virtually unchanged during the 1974 to 2012 timespan with only marginal modifications to undeveloped areas thereby fulfilling the HCDN's minimal anthropogenic disturbance criteria. As Figure 4 also highlights, the most modified regions fed the 08195000 and 08198500 gauges, both in Uvalde County, and those changes occurred sometime within the period of 1974 to 1982.

Vegetative cover, and by extension LULC, directly impact different areas of the hydrological cycle such as soil moisture, ET, runoff, and infiltration [48,65] as well as streamflow. Greater vegetative cover buffers precipitation and ultimately decreases runoff, and as a consequence, stream discharge [66]. Some brush species also impact certain hydrologic processes as they alter spring flows and stream discharge that is interconnected with aquifer systems [67]. As a result, land use changes in the NRB, while minimal, could have still impacted the hydrological dynamics in the region. Furthermore, studies by [68,69] explain that land use changes can lead to declines in stream discharge in areas that are susceptible to low flows. The minimal LULC modifications could therefore signal an increase in human consumption along the course of the different stream segments, potentially leading to decreased streamflows as less water arrives to the HCDN gauges.

The authors of [49,51,63] all agree that when stream discharge trends show results that are inconsistent with climatological variation, anthropogenic influences such as water management might be a reason for the changes. As mentioned previously, Texas' state-owned water is managed using water rights, which are granted to select users via water rights permits. The Texas Watermaster Program enforces water permits, but only in a handful of watersheds such as the Rio Grande and Brazos does the body manage the entire river basin. In the case of the NRB, some regions are under the control of the South Texas Watermaster [70], but the upper sections, which were part of our study, are not. Therefore, water rights owners exceeding their water use quotas could be another possible reason why the declines were detected by the gauges. Additionally, many water permit holders have access to the ecologically significant stream segments that were studied. Their water usage activities are concentrated upstream of the gauges and primarily involve diversion points, which are authorized locations on the channel of a stream where water can be legally removed. After closer review of the non-Watermaster records, it was confirmed that many permit owners, particularly in Uvalde and Medina Counties, are found in areas without Watermaster supervision. The impact of watershed management practices on water quality trends in rivers in the United States has also been recently pointed out by [71].

Furthermore, water used by permit holders in the non-Watermaster areas are supposed to be annually self-reported [72]. When archived reports for the previously named counties were evaluated from [72], some permittees did not report their usage during different time periods, which is mandatory even if they did not utilize their designated quotas [73]. Therefore, should permit holders who operate in non-Watermaster areas subvert the honor system and not comply with their permits, they could

potentially overuse and underreport the amount of water that they were assigned. Years of sustained overexploitation could have been reflected in the declining stream discharge, particularly during low streamflow periods, which could explain why the low flow metrics showed instances of statistically significant decreases. Water rights are prioritized based on seniority among permit owners using priority dates, so permittees with earlier water rights are given greater preference during lower flow periods [74]. Consequently, users in non-Watermaster areas could withdraw more than their permits allow, yet go unnoticed unless more senior water rights holders are impacted to the point that they file complaints for investigation.

## 5. Conclusions

In conclusion, the study used a collection of hydrologic metrics (MMK trend analyses, low flow indicators, streamflow-precipitation elasticity, as well as drought indices) on six USGS gauges in the upper NRB to robustly investigate the area's hydroclimatic relationships and the potential impacts of those associations to south Texas. The hydroclimatic analyses of the chosen stream gauges showed the relationship between the studied hydrological variables in the upper NRB existed. The assessments also detected significant decreasing stream discharge trends in a number of gauges for both the 1970 to 2014 period and during the last 20 years of the study. Although the causes of the declining streamflow trends could not be totally attributed to rainfall fluctuations, a combination of other climatological, environmental, and water management factors were assessed as probable contributors to reductions in flow. The decreasing trends should be of concern, however. The declining stream discharge could see negative impacts on the stream segments that were designated as ecologically significant, particularly for their hydrologic, biological, and ecological functions to the NRB, and by extension, south Texas. More importantly, the fact that decreasing streamflow was detected in all of the gauges at some point or the other over the study period should be of interest to the region's water and natural resource managers as the declines may have implications for current as well as future water-planning and aquatic health. It must also be noted that our findings will likely have broader impacts beyond the NRB considering that other parts of the state have stream segments designated as ecologically-significant. As indicated in [75], these include segments in Regions C, H, J, K, and L, shown in Figure 1 for spatial reference.

Continued declining stream discharge trends could directly impact the hydrologic functions of the NRB and potentially lead to negative implications for the economic, agricultural, and industrial sectors for south Texas, as the region's surface and groundwater supply heavily depends on the drainage basin. Additionally, if the streamflow that traverses the NRB down to Nueces Bay continues to decline, changes in water chemistry, disruptions to nutrient movement via nutrient spiraling, biological functions, biodiversity, natural organismal migration patterns, reproduction habits of aquatic life, as well as the overall ecological health of the region could be impaired. Further decreases in streamflow could spell even more danger for the already vulnerable endemic, threatened, and endangered populations and ecosystems of the Blanco blind salamander (*Eurycea robusta*), Comal blind salamander (*Eurycea tridentifera*), Plateau shiner (*Cyprinella lepida*), and Blanco River Springs salamander (*Eurycea pterophila*), for example, which are highlighted in [23].

As it specifically relates to the Nueces Bay, reduced stream discharge could be detrimental to its aquatic species, and by extension, the fisheries industry in that marine environment. The researchers of [31] noted that the Nueces Bay relies on the flows from the NRB to dilute the saline water to make the coastal location habitable for brackish water species. With decreasing stream discharge and rising global sea levels because of changes in climate [76] as well as thermal expansion due to global warming [77], the ability to buffer the saline conditions in the Nueces Bay would become more difficult and possibly make the coastal environment inhospitable to coastal marine life.

Furthermore, riverine plants, specifically riparian vegetation and macrophytes, play vital roles in freshwater environments and overall stream health such as, for maintaining water quality, biodiversity, spawning and breeding grounds, aquatic habitats, as well as bank stabilization [78]. With sustained

periods of unusual flow patterns such as reduced stream discharge, the diversity of riparian plant species changes from more to less water-tolerant [79] thereby impacting the important functions riverine plants provide to lotic environments and emphasizing the detrimental effects declining flows could cause.

As seen in the results and preceding content, the effects of declining stream discharge could be detrimental to numerous aspects of ecological health as well as the hydrological balance of the NRB. Therefore, it is imperative that water managers and resource planners consider strategies to mitigate the potential impacts of such effects. The possible consequences of the distribution of water rights should be evaluated and the mechanisms to emphasize the significance of truthful reporting of water use quotas, particularly since the upper regions of the NRB is in a non-Watermaster area, should be considered.

**Author Contributions:** Each author played a substantial role in the research effort. The data curation, formal analysis, and investigation were completed by E.D.T. Conceptualization, methodology, project administration, and supervision were done by K.V. The responsibility of completing the original draft and visualization fell on E.D.T. and K.V., while the project's validation was handled by V.C., N.K. and K.V. Each participant provided resources at various points of the research as well as reviewed and edited the document.

**Funding:** This research received no external funding.

**Acknowledgments:** We would like to acknowledge the insightful comments and suggestions provided by the reviewers that helped enhance the quality of this work. Tarleton State University's College of Science and Technology, particularly The Center for Environmental Studies, also deserves special mention for sponsoring E.D.T.'s graduate research.

**Conflicts of Interest:** The authors declare no conflicts of interest.

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
