# Peer review of "Hydrologic Trends in the Upper Nueces River Basin of Texas—Implications for Water Resource Management and Ecological Health"

_hydrology, doi:10.3390/hydrology6010020_

Round 1

Reviewer 1 Report

I found this paper to be generally well written. The language is clear, the logic is easy to follow and the literature is current. The methods used are conventional and, as far as I can tell, have been applied correctly. I have a few corrections in the text. In some places, the authors use “etc” at the end of a list. This is not conventional and should be deleted. The use of units should also be more consistent. They begin with imperial units but then toward the end convert to metric.

Here are some other suggested changes.

Table 2, ln.212. This table is difficult to read. It would be better if the data were presented as plots. Then, readers would be able to infer some trends even from the plots.

Line 362. The Q0, Q50, and Q100 notation is confusing. Change to Qmin, Qmed and Qmax, which is really what these quantities are.

Ln 386. The rationale for the analysis of trends from 1994-2004 seems unconvincing. It appears as if the authors are looking for data segments that will give them the results they want to see. It can not be that temperatures in Texas began to rise only after 1994. If you must analyze the data for this or other subperiod, then there needs to be a reason in the data e.g. an abrupt change point that shows this period to be distinct in some way. Otherwise, the choice seems arbitrary.

Section 3.5 Spearman’s correlation

This section is not necessary. Everyone knows that precipitation and streamflow have some correlation albeit sometimes difficult to demonstrate. Part of the challenge here is that you are taking a point based data (precipitation) and trying to correlation it with river discharge which is an areally integrated variable. It is questionable how meaningful this correlation is or what purpose it serves. It would make more sense to have an areally integrated precipitation estimate to correlate against the river discharge data.

Lin 430, Table 7. Show the p-values associated with these correlations on the table.

The major concern I have with the manuscript is the attempts to make associations that are not clear or may be even non-existent. Consider the claim beginning on line 468. The authors state that although statistically significant trends were uncommon… a relationship exists!! Why are you trying to find a relationship if you have not been able to demonstrate one? Your analysis either shows a relationship or it doesn’t. If it doesn’t, there could be good reasons why not and this is what you should try to discover/highlight. If it does, simply report it. You should not infer a relationship based on circumstantial data! This part and similar paragraphs must be changed if not deleted entirely.

p.473. You make reference to extreme drought here but this was never defined.

Author Response

Comments and Suggestions for Authors

1. “I found this paper to be generally well written. The language is clear, the logic is easy to follow and the literature is current. The methods used are conventional and, as far as I can tell, have been applied correctly. I have a few corrections in the text. In some places, the authors use “etc” at the end of a list. This is not conventional and should be deleted. The use of units should also be more consistent. They begin with imperial units but then toward the end convert to metric.”

Authors’ response: We thank the reviewer for the encouraging words. All references to the abbreviation “etc” were removed, and only imperial units are used throughout the revised manuscript.

Other suggested changes

2. “Table 2, ln.212. This table is difficult to read. It would be better if the data were presented as plots. Then, readers would be able to infer some trends even from the plots.”

Authors’ response: In response to the reviewer’s suggestion, we have presented the daily flow data in the form of a violin plot (Figure 5). A brief explanation of the figure has also been added in line # 197-205 in the revised manuscript for the benefit of readers who may not be familiar with the format of presentation.   

3. “Line 362. The Q0, Q50, and Q100 notation is confusing. Change to Qmin, Qmed and Qmax, which is really what these quantities are.”

Authors’ response: All uses of the notations Q0, Q50, and Q100 were removed and replaced with the suggested abbreviations Qmin, Qmed and Qmax respectively.

4. “Ln 386. The rationale for the analysis of trends from 1994-2004 seems unconvincing. It appears as if the authors are looking for data segments that will give them the results they want to see. It [cannot] be that temperatures in Texas began to rise only after 1994. If you must analyze the data for this or other subperiod, then there needs to be a reason in the data e.g. an abrupt change point that shows this period to be distinct in some way. Otherwise, the choice seems arbitrary.”

Authors’ response: Trends in mean annual temperatures in South Texas between 1930 and 2001 were studied by Norwine and John (2007) [The Changing Climate of South Texas, 1900-2100: problems and prospects, impacts and implications.2007]. These authors fit regression equations for temperature based on data collected from 16 stations across the study region, which generally overlaps the NRB. Their findings suggest an upward trend in temperature around the mid-1990s which led us to demarcate the 1994-2014 period for separate analysis. The shorter 1994-2014 period is first mentioned in Section 2.3.1 and therefore we have included this statement for clarification in line # 288-293 in the revised manuscript.  

Section 3.5 Spearman’s correlation

5. “This section is not necessary. Everyone knows that precipitation and streamflow have some correlation albeit sometimes difficult to demonstrate. Part of the challenge here is that you are taking a point based data (precipitation) and trying to correlation it with river discharge which is an areally integrated variable. It is questionable how meaningful this correlation is or what purpose it serves. It would make more sense to have an areally integrated precipitation estimate to correlate against the river discharge data.”

Authors’ response: Based on the reviewer’s comment about this section being unnecessary, we have removed it from the revised manuscript. This change also necessitated the removal of Section 2.3.4 from the original manuscript where correlation was first introduced as well as Table 7, where correlation results were presented.  

6. “[Ln] 430, Table 7. Show the p-values associated with these correlations on the table.”

Authors’ response: Based on the reviewer’s earlier comment, the entire discussion on correlations was removed from the manuscript. Therefore, p-values are not shown in the revised manuscript.

7. “The major concern I have with the manuscript is the attempts to make associations that are not clear or may be even non-existent. Consider the claim beginning on line 468. The authors state that although statistically significant trends were uncommon… a relationship exists!! Why are you trying to find a relationship if you have not been able to demonstrate one? Your analysis either shows a relationship or it doesn’t. If it doesn’t, there could be good reasons why not and this is what you should try to discover/highlight. If it does, simply report it. You should not infer a relationship based on circumstantial data! This part and similar paragraphs must be changed if not deleted entirely.”

Authors’ response: We agree with the reviewer that some statements in the discussion contradict each other. In response to the reviewer’s suggestions, certain sections from the discussion were deleted without sacrificing the key results obtained in this study and the continuity of the manuscript. These include:

[From the end of the first paragraph of Section 4] “Furthermore, although statistically significant precipitation trends were uncommon, the results from the drought indices, streamflow-precipitation elasticity, and Spearman’s Rho demonstrated that a relationship between the hydroclimatic variables exists, which suggests that changes in precipitation resulted in variations in stream discharge on some level.” This statement appeared in line # 454-458 of the original manuscript and is now deleted.  

[From the second paragraph of Section 4 (the claim the reviewer was referring to)] “The precipitation trend results could possibly explain why significant drops in streamflow were identified during the consecutive fall and winter seasons, particularly for the central gauges found in Uvalde County over the 1970 to 2014 timespan.” This statement appeared in line # 467-470 of the original manuscript and is now deleted.

8. “p.473. You make reference to extreme drought here but this was never defined.”

Authors’ response: The term “extreme drought” originally appears in Section 3.6. All references to “extreme drought” have now been removed from the manuscript. Instead, the term “drought” as defined as an event that has an index value <-1, is defined and used in line # 412 of the revised document.

Reviewer 2 Report

The authors use a data set, which covers the period from 1970 to 2014, in order to assess the hydroclimatic trends of the Nueces River Basin and their impacts to the recharge of the Edwards Aquifer, which has a crucial role in the area. The methods that the authors use are: modified Mann-Kendall test (MMK), annual minimum and 7-day low flow indicators, streamflow elasticity, Spearman’s Rho correlation, and drought indices [Streamflow Drought Index (SDI) and Standardized Precipitation Index (SPI)]. The results of their work are very detailed and well presented. The discussion part also include useful information, but mainly focus on a limited audience.

The submitted paper is a regional case study research, with almost all the references to be focused in the area. I could imagine that the target group of the paper is regional, however a research paper characterized as “article” should cover the demands of a bigger audience. My main obstacle to fully support the proposed work, is that also the discussion part as well as the conclusion part solely focus to the case study area.  In order to facilitate this opening to the broader scientific community, the adding of universal literature is a good solution.

E.g. In line 67, the authors could add that “similar approaches to regional water planning regions, are also met in the European Union with the implementation of the River Basin Management Plans (Skoulikaris, C. & Zafirakou, A. River Basin Management Plans as a tool for sustainable transboundary river basins’ management. Environ Sci Pollut Res (2019). https://doi.org/10.1007/s11356-019-04122-4”

Eg. In lines 582-584, the authors could add similar behaviours in their literature, (Murphy, J., Sprague, L. 2019. Water-quality trends in US rivers: Exploring effects from streamflow trends and changes in watershed management. Science of the Total Environment, 656, pp. 645-658.

Finally, the English are excellent, but the document needs to be partially restructured.

I propose major revisions only because of my comment no. 5, otherwise the paper would have been proposed for minor revisions.

 General comments

The introduction part of the manuscript is too extensive and too detailed with regional information. Consequently, the readers might lose their interest aim on the manuscript. Thus I would propose the restructure of the introduction part, as well as the reduction of the length of some paragraphs. Please see the Specific comments 2, 3, 4.

Specific comments

Line 23:” …which impacts streamflow reliability”. The streamflow has not a reliability. I suppose the authors would like to say something like streamflow continuity? At any case please rephrase.

Lines 95-104: The authors provide too much information that does not support their research. I would propose this paragraph to be shorten, let’s say to 2 lines, just indicating the environmental importance of the specific area.

Lines 113-143: This part of the introduction I would propose to be transferred to the methodological part, by adding a new section e.g. entitled: Background information.

Lines 151-154: This sentence could be transferred to the discussion part of the paper.

Lines 215-217: “To accurately represent precipitation contributions to a watershed, precipitation stations nearest to streamflow gauging sites should be chosen”. This approach is completely wrong. The precipitation cannot be distinguished to the one that is close to a station and to the one that is not. The precipitation that falls in a subbasin is responsible for the creation of the streamflow, with the rain that falls close to the stream to enter faster into the stream, while the rain that falls in the upper part of the subbasin enters into the main stream with a time lag. The authors should revise this part of their manuscript in conjunction to the theory of concentration time.  (new calculations might be necessary)

Author Response

Comments and Suggestions for Authors

“The authors use a data set, which covers the period from 1970 to 2014, in order to assess the hydroclimatic trends of the Nueces River Basin and their impacts to the recharge of the Edwards Aquifer, which has a crucial role in the area. The methods that the authors use are: modified Mann-Kendall test (MMK), annual minimum and 7-day low flow indicators, streamflow elasticity, Spearman’s Rho correlation, and drought indices [Streamflow Drought Index (SDI) and Standardized Precipitation Index (SPI)]. The results of their work are very detailed and well presented. The discussion part also includes useful information, but mainly [focuses] on a limited audience.”

“The submitted paper is a regional case study research, with almost all the references to be focused in the area. I could imagine that the target group of the paper is regional, however a research paper characterized as “article” should cover the demands of a bigger audience. My main obstacle to fully support the proposed work, is that also the discussion part as well as the conclusion part solely focus to the case study area. In order to facilitate this opening to the broader scientific community, the adding of universal literature is a good solution.”

Authors’ response: We thank the reviewer for graciously helping us with the requisite literature needed to reach the broader audience of the journal.

“E.g. In line 67, the authors could add that “similar approaches to regional water planning regions, are also met in the European Union with the implementation of the River Basin Management Plans (Skoulikaris, C. & Zafirakou, A. River Basin Management Plans as a tool for sustainable transboundary river basins’ management. Environ Sci Pollut Res (2019). https://doi.org/10.1007/s11356-019-04122-4

Eg. In lines 582-584, the authors could add similar behaviours in their literature, (Murphy, J., Sprague, L. 2019. Water-quality trends in US rivers: Exploring effects from streamflow trends and changes in watershed management. Science of the Total Environment, 656, pp. 645-658.”

Authors’ response: We have added these new references to the revised document; they appear in line # 68-70 and line # 530-531, respectively.

Additionally, we have made references to the ecological segment designation in other parts of Texas, managers of which may find our results interesting and useful. This addition can be found in line # 560-563 of the revised manuscript.  

“Finally, the English [is] excellent, but the document needs to be partially restructured.

I propose major revisions only because of my comment no. 5, otherwise the paper would have been proposed for minor revisions.”

General comments

1. “The introduction part of the manuscript is too extensive and too detailed with regional information. Consequently, the readers might lose their interest aim on the manuscript. Thus I would propose the restructure of the introduction part, as well as the reduction of the length of some paragraphs. Please see the Specific comments 2, 3, 4.”

Authors’ response: Please see the specific responses to these individual comments below.

Specific comments

2. “Line 23:” . . . which impacts streamflow reliability”. The streamflow has not a reliability. I suppose the authors would like to say something like streamflow continuity? At any case please rephrase.”

Authors’ response: We have deleted the word reliability in the abstract.

3. “Lines 95-104: The authors provide too much information that does not support their research. I would propose this paragraph to be [shortened], let’s say to 2 lines, just indicating the environmental importance of the specific area.”

Authors’ response: We have retained and moved the statement about the basin’s ecological significance to the previous paragraph (appears on line # 95-99 in the revised manuscript) and deleted the remainder of the original paragraph.

4. “Lines 113-143: This part of the introduction I would propose to be transferred to the methodological part, by adding a new section e.g. entitled: Background information.”

Authors’ response: We have moved this paragraph as a separate section (2.1) titled Background Information. This new section can be found in the beginning line # 123 in the revised manuscript. However, this move required the deletion of the last statement in the previous paragraph “A brief review of the existing literature follows”.

5. “Lines 151-154: This sentence could be transferred to the discussion part of the paper.”

Authors’ response: As per the reviewer’s suggestion, we have moved the statements about the novel aspects of our study to the very beginning of the discussion section; they appear in line # 438-444 in the revised manuscript.

6. “Lines 215-217: “To accurately represent precipitation contributions to a watershed, precipitation stations nearest to streamflow gauging sites should be chosen”. This approach is completely wrong. The precipitation cannot be distinguished to the one that is close to a station and to the one that is not. The precipitation that falls in a subbasin is responsible for the creation of the streamflow, with the rain that falls close to the stream to enter faster into the stream, while the rain that falls in the upper part of the subbasin enters into the main stream with a time lag. The authors should revise this part of their manuscript in conjunction to the theory of concentration time. (new calculations might be necessary)”

Authors’ response: The choice of precipitation gauge stations used in the study was made entirely on the basis of availability of serially-continuous data. Some stations had near-complete records for the 45-year period of analysis but there were occasional gaps in the series. These gaps were filled in using spatial interpolation of synchronous observations from adjacent stations. Other stations did not have appreciable completeness or continuity – data from multiple stations had to be consolidated in these cases. Therefore, the rationale here is the completeness of the precipitation data (without gaps) rather than the nearness of precipitation station to the discharge measurement location. We have removed the misleading text. This section has also been modified to allow for continuity. These changes appear in line # 216-226 of the revised manuscript. We have also renamed Table 2 from “Representative precipitation stations” to “Consolidated precipitation stations”.  

Round 2

Reviewer 2 Report

The authors improved their document according to all the reviewers comments. Thus, the final result is a scientifically sound manuscript that I proposed to be accepted for publication.